# Modelling the impacts of an invasive species across landscapes: a step-wise approach

Darren Ward and Fraser Morgan

Landcare Research, Auckland, New Zealand

## ABSTRACT

We estimate the extent of ecological impacts of the invasive Asian paper wasp across different landscapes in New Zealand. We used: (i) a baseline distribution layer (modelled via MaxEnt); (ii) Asian paper wasp nest density (from >460 field plots, related to their preferences for specific land cover categories); and (iii) and their foraging intensity (rates of foraging success, and the time available to forage on a seasonal basis). Using geographic information systems this information is combined and modelled across different landscapes in New Zealand in a step-wise selection process. The highest densities of Asian paper wasps were in herbaceous saline vegetation, followed closely by built-up areas, and then scrub and shrubland. Nest densities of 34 per ha, and occupancy rates of 0.27 were recorded for herbaceous saline vegetation habitats. However, the extent of impacts of the Asian paper wasp remains relatively restricted because of narrow climate tolerances and spatial restriction of preferred habitats. A step-wise process based on geographic information systems and species distribution models, in combination with factors such as distribution, density, and predation, create a useful tool that allows the extent of impacts of invasive species to be assessed across large spatial scales. These models will be useful for conservation managers as they provide easy visual interpretation of results, and can help prioritise where direct conservation action or control of the invader are required.

## INTRODUCTION

Species distribution model, which can be used to predict a species' potential occurrence across a landscape, have become a key part of ecological research and conservation planning (*Guisan & Thuiller, 2005*; *Thuiller et al., 2008*; *Elith & Leathwick, 2009*; *Franklin, 2013*). Such models can explore more than just distribution, and are increasingly being used for a range of biodiversity applications such as modelling the distribution of communities, ecological refuges, potential impacts under climate change, and biotic interactions (*Araújo & Luoto, 2007*; *Bradley, 2013*; *Porto, Carnaval & da Rocha, 2013*; *Ross & Howell, 2013*).

One aspect of species distribution models that are less utilised is analyses of the impacts of invasive species across large spatial scales. Yet species distribution models are ideal for this type of study because they capture localised impacts (at sites) and can extend these

Corresponding author
Darren Ward,
wardda@landcareresearch.co.nz

impacts across larger spatial scales using geographic information systems. Predicting impacts across landscapes is an extremely useful tool for invasive species, and may highlight the need for direct conservation action or control of the invader.

Social insects form a large part of the invasive invertebrate literature because they usually have a wide host range, feed at a range of trophic levels, can reach very high densities, often have noticeable effects on prey, and are commonly associated with human trade (*Suarez, Holway & Ward, 2005*; *Snyder & Evans, 2006*; *Wilson, Mullen & Holway, 2009*; *Ward et al., 2010*; *Roura-Pascual et al., 2011*; *Roy et al., 2011*).

Paper wasps (Hymenoptera: Vespidae) are widely distributed around most of the globe and are diverse and common in many landscapes (*Beggs et al., 2011*). Paper wasps are likely to influence many other species in terrestrial ecosystems because they are voracious predators of invertebrates (*Ward & Ramón-Laca, 2014*). Four species of paper wasps are invasive around the globe (*Beggs et al., 2011*): *Polistes versicolor* (Olivier) in the Galápagos; *P. dominula* in North America; and *P. humilis* (Fabricius) and *P. chinensis antennalis* Perez in New Zealand.

In New Zealand, *P. chinensis antennalis*, commonly known as the Asian paper wasp, was first recorded in 1979, and has subsequently spread rapidly across much of the North Island and several locations in the South Island (*Clapperton, Tilley & Pierce, 1996*). Although the distribution and nesting biology of the Asian paper wasp is well known (*Clapperton, Tilley & Pierce, 1996*; *Yamane, 1996*; *Clapperton & Dymock, 1997*; *Clapperton & Lo, 2000*), little has been published on its ecological impacts. Densities of paper wasps can reach up to 210 nests/ha (*Clapperton, 1999*), and although densities of 20–40 nests/ha are more common, this still translates into ∼1,000–2,000 wasps/ha, who are responsible for many 10,000s of prey captured per season (*Clapperton, 1999*).

Paper wasps do not naturally occur in New Zealand (along with many other groups of social insects (*Ward et al., 2006*)), and consequently, there are concerns that the Asian paper wasp could have a significant impact on native biodiversity, particularly on the larvae of butterfly and moths, which are heavily preyed upon (*Clapperton, 1999*; *Ward & Ramón-Laca, 2014*). In this paper we used a potential distribution layer (modelled via MaxEnt), and added land-cover-specific densities, and foraging intensity in a step-wise process, to develop a comprehensive model of the spatial extent of the impacts of Asian paper wasps across New Zealand.

## MATERIALS AND METHODS

### Distribution

#### *Occurrence data*

In total, 253 geo-referenced occurrence records of the presence of Asian Paper wasps were obtained from (a) published literature (*Clapperton, 1999*) ($n = 71$); (b) museum collections ($n = 112$); (c) field surveys in 2012 ($n = 22$); and (d) through a publicity campaign asking members of the public for sightings of the Asian paper wasp ($n = 48$; only records that were supported by photographic evidence were used, e.g., a worker wasp, or

nest, both of which are very distinct). Specimens from museum collections are held in the Auckland War Memorial Museum (Auckland), the New Zealand Arthropod Collection (Auckland), Te Papa (Wellington), Entomology Research Museum (Lincoln University), and Otago Museum (Dunedin).

### Environmental variables

Environmental layers used were elevation (m), degree days (at a 10 °C base, where degree days are calculated as the average daily temperature minus the temperature base of 10 °C, and is accumulated over the course of a year), annual rainfall (mm), solar radiation ($MJm^{-2} day^{-1}$), maximum annual temperature (°C), and minimum annual temperature (°C). Environmental layers came from the National Institute of Water and Atmospheric Research, except for elevation which is available from Landcare Research LRIS portal (http://lris.scinfo.org.nz/). Each environmental layer was created at 500 m (25 ha) spatial resolution. The nesting biology of paper wasps is known to be strongly influenced by such abiotic variables (*Yamane, 1996*).

### Model method

Distribution models were generated using MaxEnt (Version 3.3.3) to discriminate the environments associated with the presence of Asian paper wasps from the rest of the landscape (*Phillips, Anderson & Schapire, 2006*; *Elith et al., 2011*). MaxEnt software uses the principle of maximum entropy to relate distribution records of a species to environmental variables in order to estimate a species' potential geographical distribution (*Phillips, Anderson & Schapire, 2006*). It is a well-studied method considered to produce robust results with sparse or irregularly sampled data, which is often the case with poorly known species (*Elith et al., 2011*). It is freely available at http://www.cs.princeton.edu/∼schapire/maxent/. MaxEnt models were trained with a random sample of 75% of the species occurrence data, and the remaining 25% was used to test model performance (*Guisan & Zimmermann, 2000*). We used 50 model runs on random subsamples of the occurrence data to assess uncertainty of the species distribution models predictions. We used the 'area under the curve' (AUC) as single measure of overall model accuracy that is not dependent upon a particular model threshold (*Fielding & Bell, 1997*). The MaxEnt output is a logistic probability with values between 0 (low probability) and 1 (high probability).

As the occurrence dataset was not constructed using a systematic sampling approach, a geographic sampling bias may occur (*Phillips et al., 2009*; *Syfert, Smith & Coomes, 2013*). Therefore, we created a sampling bias grid in MaxEnt using a quartic kernel density layer (Fig. 1) to correct for this bias as recommended (*Phillips et al., 2009*; *Syfert, Smith & Coomes, 2013*). To provide a surface that highlights both suitable and unsuitable habitats, we require a threshold value that outlines a minimum value that constitutes a suitable habitat. After reviewing the continuous raster output and the nominal threshold values from the MaxEnt models, we selected the average 10% minimum threshold for all 50 MaxEnt runs to define the minimum probability of a suitable habitat. This value was selected as it provides flexibility to account for the variation in quality of the input data.

**Peer**J

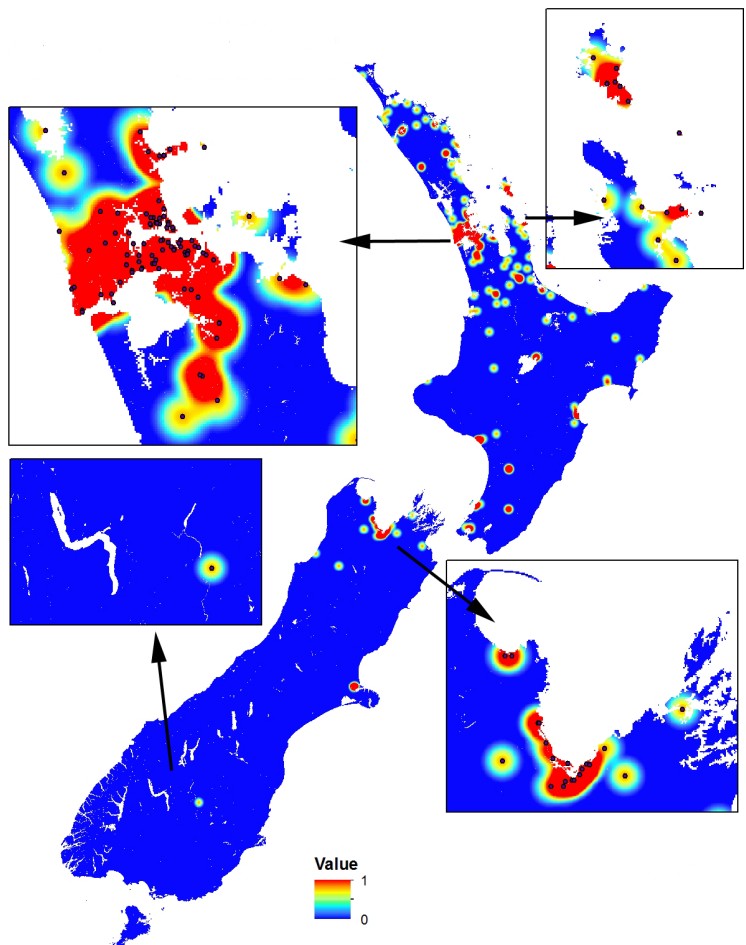

**Figure 1 Sampling bias grid in MaxEnt using a quartic kernel density to correct for geographic sampling bias in occurrence data.** The scale is a logistic probability with values between 0 (low probability; blue) and 1 (high probability; orange-red). Inset boxes visually show the sampling density kernel for different regions.

## Density

We used previous information on habitat preferences of the Asian paper wasp in combination with land cover satellite imagery to undertake field surveys to generate density estimates and occupancy rates of the Asian paper wasp. We then combined these land cover-specific densities with the MaxEnt distribution model to generate density estimates of Asian paper wasps across New Zealand.

### Land cover

Asian paper wasps occur across a range of native 'open-canopy-habitats' such as grassland, marshland and shrubland, where nests are typically found within short (<2 m high) plants (*Clapperton, 1999*; *Clapperton & Lo, 2000*). They are also common in urban areas, with nests found in residential gardens and on wooden fences. Although the habitat preferences

**Table 1** Estimates of the total area occupied and total number of nests for the Asian Paper Wasp extrapolated across New Zealand from field surveys of density and occupancy for specific land cover classes.

| Land cover class | Total potential (ha) | Plots | Nests | Occupancy | Total occupied (ha) | Average density | Total nests |
|---|---|---|---|---|---|---|---|
| Herbaceous saline vegetation | 13,050 | 114 | 39 | 0.27 | 3,523 | 34 | 119,799 |
| Built-up area | 146,275 | 93 | 27 | 0.23 | 33,643 | 29 | 975,654 |
| Scrub and shrubland | 567,100 | 206 | 18 | 0.09 | 51,039 | 9 | 459,351 |
| Orchard vineyard & other perennial crops | 100,575 | 0 | | 0.03[*] | 3,017 | 3[*] | 9,051 |
| Herbaceous freshwater vegetation | 47,125 | 0 | | 0.03[*] | 1,413 | 3[*] | 4,241 |
| Tall tussock grassland | 250 | 0 | | 0.03[*] | 7 | 3[*] | 22 |
| Forest | 0 | 53 | 0 | 0 | 0 | 0 | 0 |
| Total | 874,375 | 466 | 84 | 0.15 | 92,642 | 18 | 1,568,118 |

**Notes.**

[*] Nominal values (see methods).

of Asian paper wasps are well established, there is little information on their densities within these habitats.

In order to select habitats for field surveys to generate density estimates, we used the LCDB-3 database (*LCDB NZ Land Cover Database, 2012*) derived from the 2007–2008 LUCAS satellite imagery, which classifies land cover into 33 classes. Three land cover classes are highly suitable for Asian paper wasps: (i) built-up area; (ii) herbaceous saline vegetation; and (iii) scrub and shrubland.

Some land cover classes were not surveyed because they are not suitable for Asian paper wasps (e.g., water bodies; bare or lightly vegetated surfaces; artificial surface (such as roads)); are too disturbed to allow nests to develop (e.g., cropland; pasture); or are not preferred (e.g., forest types, alpine vegetation) (*Clapperton, 1999*). We considered some land cover classes (orchard vineyard & other perennial crops; herbaceous freshwater vegetation; tall tussock grassland) to be possibly suitable but of very low preference (due to high disturbance and unsuitable vegetation to construct a nest), and did not survey these classes but instead estimated a nominal value of three nests per ha and occupancy rate of 0.03.

### Field surveys

During January to March 2012 we surveyed the three most suitable land cover classes: built-up area; herbaceous saline vegetation; scrub and shrubland. We walked slowly through plots ($10 \times 10$ m) checking vegetation for nests of Asian paper wasps (Table 1). We believe we achieved a high detection rate as nests are larger and more obvious from January to March. However, as it is possible we failed to detect all nests, our densities are conservative. All plots were in the Auckland region.

## Impacts on prey

Paper wasps are generalist predators of invertebrates, and we inferred their effects on biodiversity through foraging intensity as measured by: (i) their foraging success of prey capture, and (ii) the time available to forage.

### Foraging success rate

The foraging of Asian paper wasps has been studied at three sites in northern New Zealand (*Clapperton, 1999*). We used some of this published data to derive a foraging success rate based on: (i) the return of foraging wasps ("traffic rate"; average 0.33 per minute; range 0.23–0.46); (ii) the proportion of wasps that carried material back to the nest (average 0.30; range 0.25–0.38), and (iii) the proportion of material that represented a prey item (as opposed to liquid food or nesting material) (average 0.75; range 0.70–0.80, excluding the value 0.12 as an outlier). Multiplying these data gives a foraging success rate of 4.5 prey captured/nest/hour (range 2.4–8.4).

### Foraging time available

Available foraging time to AWP was estimated by examining "sunshine hours" across New Zealand. Paper wasps do not forage at night or in periods of rain (*Clapperton, 1999*). Sunshine hours accounts for periods of cloud cover and rain, which reduce foraging, but are also topographically very accurate, and take into consideration the landscape effects of slope and gullies, etc. We used hourly sunshine data ($MJm^{-2}$, obtained from the National Institute of Water and Atmospheric Research, 500 m spatial resolution), and summed these across days, months, and the period February to April (when paper wasps are active).

Foraging success and the total time available to forage (for the period February to April, when paper wasps are active) was multiplied together, and then combined with land cover specific densities and the MaxEnt distribution model to develop a comprehensive model of the spatial extent of the impacts of Asian paper wasps across New Zealand.

## RESULTS

### Distribution

The MaxEnt model predicted that coastal and lowland regions of the North Island are highly suitable for the Asian paper wasp, with potential to extend inland and inhabit considerable areas of the middle and lower North Island (Figs. 2 and 3). Suitable sites in the South Island are largely restricted to northern regions and the eastern lowland. However, coastal areas of the West Coast and Central Otago (where the Asian paper wasp has been present in Alexandra for over a decade) are also predicted to be suitable (Figs. 2 and 3). The average test AUC of the MaxEnt models was 0.846 ($\pm$0.013).

These distributions correspond very well to an intolerance of cooler mountainous and wet regions. Jackknife tests of variable importance showed elevation contributed the highest gain, and was the variable containing the most information by itself (possibly because elevation is strongly correlated with temperature and rainfall). Solar radiation was the variable that decreased the gain the most when it is omitted, and contains the most

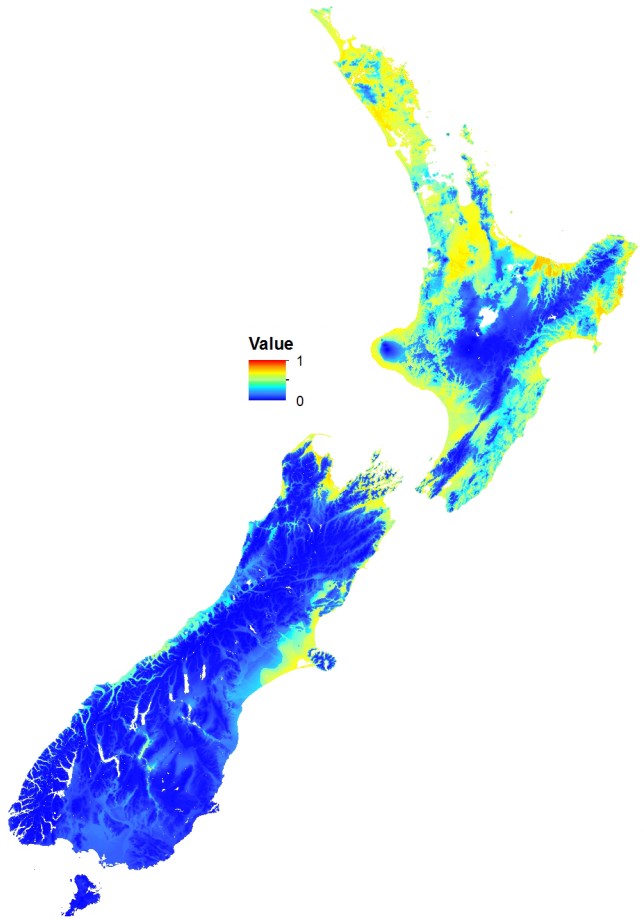

**Figure 2 Potential distribution of the Asian Paper Wasp using MaxEnt.** The scale is a logistic probability with values between 0 (low probability; blue) and 1 (high probability; orange-red).

information that is not present in other variables. Contribution to the MaxEnt model gain ranked solar radiation (43.5%) as the most important variable, followed by degree days (26.1%) and elevation (21.5%).

### Density

A total of 466 plots were surveyed, where average densities of AWP nests ranged from 0 to 34 nests per ha (Table 1). The highest densities were in herbaceous saline vegetation (HSV), followed closely by built-up areas and then scrub and shrubland (Kruskal–Wallis, $H = 21.53$, d.f. $= 2$, $P < 0.001$, Table 1). No nests were found in forest plots. A high proportion of plots had no nests (Fig. 4), thus the occupancy rate of Asian paper wasps in plots was generally low, with only 15% of plots occupied by at least one nest. However, occupancy was higher in herbaceous saline vegetation (0.27) and built-up areas (0.23), compared with scrub and shrubland (0.09). Three nests was the maximum found in a plot ($n = 2$ plots).

**Peer**J

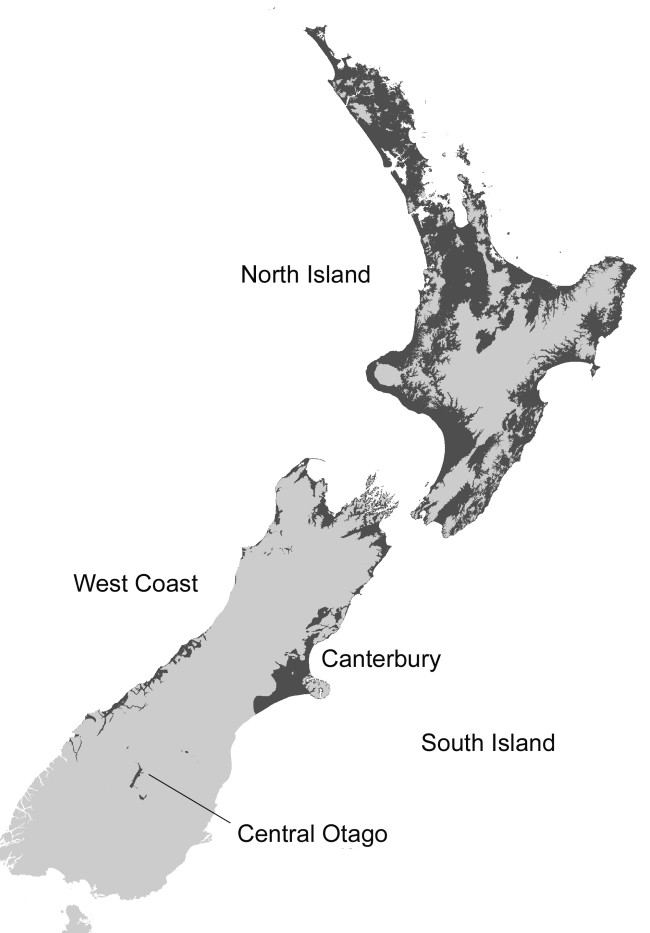

**Figure 3 Potential distribution of the Asian Paper Wasp (based on a 10% threshold in MaxEnt).** Unsuitable (light), suitable (dark).

Extrapolation of suitable land cover across New Zealand shows over 874,000 ha is estimated to be suitable for Asian paper wasps (Table 1). However, this is greatly reduced (to 92,000 ha) when rates of occupancy are included. A combination of density and occupancy data estimates the total number of Asian paper wasp nests in New Zealand in the region of 1.5 million per year (Table 1).

## Impacts on prey

Based on the estimated total number of nests across the entire country (1,568,118, Table 1), and the estimates of foraging success and foraging time available (Table 2), the Asian paper wasp is responsible for consuming an estimated 3–4 billion prey items during a single season. Total sunshine hours from different locations throughout New Zealand for the February–April period were very similar (Table 2). Consequently, there was little difference in terms of the number of prey items consumed from different locations around New Zealand (Fig. 5).

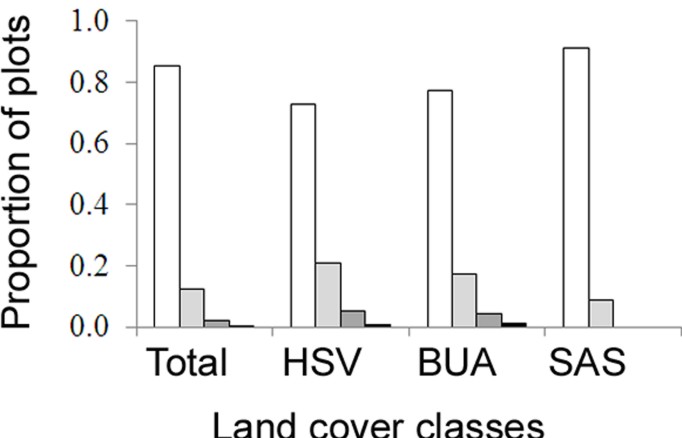

**Figure 4** **The proportion of field survey plots with zero (white), one (light grey), two (dark grey), three (black) nests.** Land cover classes are: herbaceous saline vegetation (HSV), built-up areas (BUA), and scrub and shrubland (SAS).

**Table 2** **Total sunshine hours for different locations throughout New Zealand.**

| Latitude (S) | Location | Annual | Feb/Apr season | Within potential distribution |
|---|---|---|---|---|
| −36.85 | Auckland | 1949 | 519 | Yes |
| −37.77 | Hamilton | 1954 | 529 | Yes |
| −37.67 | Tauranga | 2169 | 576 | Yes |
| −39.49 | Napier | 2161 | 550 | Yes |
| −40.36 | Palmerston North | 1852 | 526 | Yes |
| −41.30 | Wellington | 1986 | 553 | Yes |
| −43.47 | Christchurch | 2040 | 528 | Yes |
| −45.02 | Queenstown | 1927 | 545 | Outside? |
| −45.86 | Dunedin | 1594 | 407 | Outside |
| −46.41 | Invercargill | 1649 | 421 | Outside |

## DISCUSSION

Site-based studies provide important details about the impacts of an invasive species; however, these are often very limited in spatial scale. Because of this limitation it remains unknown whether the stated impacts also occur at other sites (or habitats, etc.). This is particularly true in areas that have different abiotic conditions that could directly affect the biology of an invasive species.

Creating a step-wise approach based on key features of an invasive species (distribution, density, foraging intensity) and that incorporates species distribution modelling and geographic information systems allows the extent of impacts to be examined across large spatial scales. The ability to "scale-up" impacts across large spatial scales could be extremely useful for pest management, particularly to provide a greater assessment of the
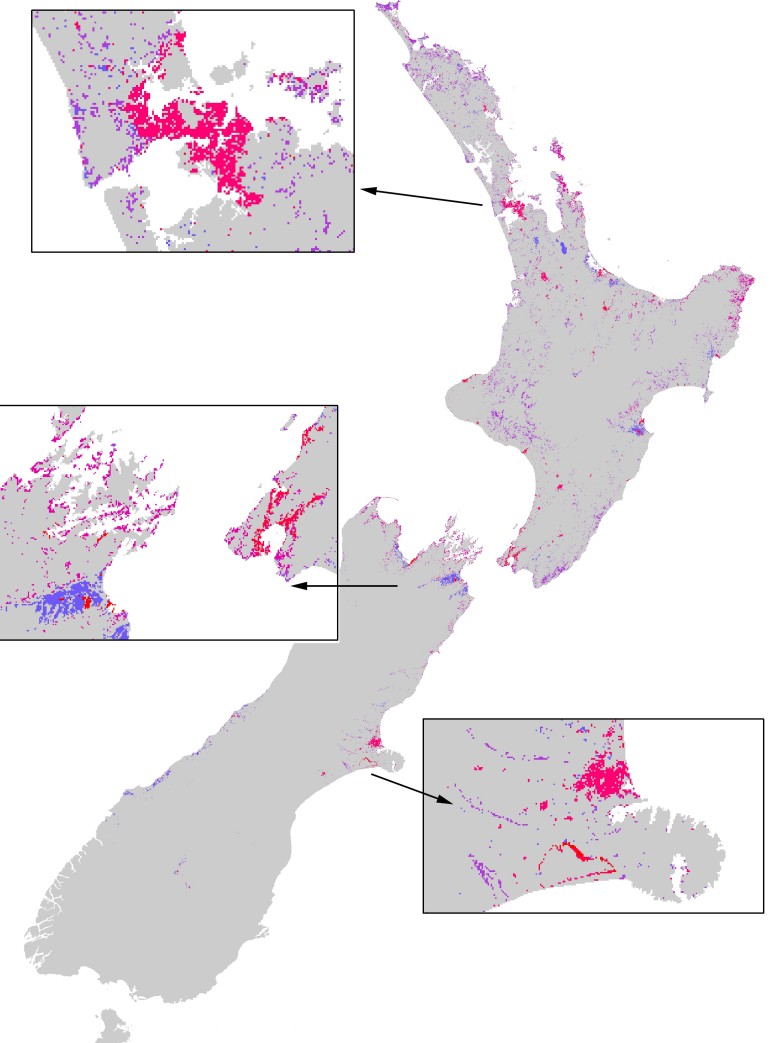

**Figure 5 The extent of the impacts of the Asian Paper wasp across New Zealand based on a combination of potential distribution, land-cover density, and foraging intensity.** The colours represent a relative scale of impacts (based on the number of prey items consumed) from grey (no impact) to blue-purple (lower impacts) to pink-red (highest impacts). Inset boxes visually show the extent of impacts for different regions.

possible impacts of an invasive species in the early stages of its invasion, before it reaches equilibrium.

Despite forming a large part of the exotic fauna worldwide, the impacts of invasive invertebrates have received disproportionately less attention compared with the impacts of plants and vertebrates, especially for impacts associated with natural ecosystems (*Kenis et al., 2009*; *Roy et al., 2011*). Although the threat posed by invasive invertebrates towards natural ecosystems is well recognised, evidence is scarce and limited to a few well-known examples (*Kenis et al., 2009*; *Brockerhoff et al., 2010*). However, New Zealand is well known

**Peer**J

for understanding the impacts of invasive invertebrates, particularly ants and social wasps in natural ecosystems (*Ward, 2007*; *Beggs et al., 2011*). The impacts of paper wasps are also of concern, especially because the native invertebrate fauna did not evolve alongside a diverse and abundant social insect fauna, and thus could be particularly susceptible to such predators. The potential impacts of such highly predacious and generalist arthropods have recently been highlighted (*Snyder & Evans, 2006*).

This paper provides an overall assessment of where the impacts of paper wasps will be, and their relative impacts between different habitats and regions. This information could ultimately direct where pest management actions should be taken. Although consuming an estimated 3–4 billion prey per year, the overwhelming impression of Asian paper wasps across New Zealand is that the extent of their impacts is very restricted (e.g., Fig. 5). A large proportion of the country is climatically unsuitable for their establishment, and because of their strong habitat preferences they are then further restricted. However, localised, or habitat-specific impacts, may be considerable. Recent research shows the Asian paper wasp prey on a large diversity of endemic caterpillars in the herbaceous saline vegetation habitat (*Ward & Ramón-Laca, 2014*). Further assessment of the impacts of Asian paper wasps should be directed towards herbaceous saline vegetation habitat because it had the highest nest density and occupancy rates.

Several aspects could be further examined to improve modelling. In particular, there is some uncertainty with the potential distribution in the Central Otago region (e.g., Queenstown), where their presence has been recorded for over a decade (*Harris, 2002*). More locality records from Central Otago would help reduce this uncertainty. Determining density and occupancy rates from other regions around New Zealand would also be valuable. The current values are based on field plots around Auckland (upper North Island). It is possible that density and occupancy rates may be less in other regions because Asian paper wasps have (and are) spreading southward and these regions are less likely to be at equilibrium for density and occupancy. Additionally, improved estimates of how abiotic factors interact with nest and foraging activity are also important to understand the rates of predation by the Asian paper wasp.

## CONCLUSION

A step-wise approach based on geographic information systems and species distribution models, in combination with factors such as distribution, density, and specific impacts on biodiversity (in this case predation) create a useful tool that allows the extent of impacts of an invasive species across large spatial scales to be assessed. These models will be useful for conservation managers as they provide easy visual interpretation of results, and can help prioritise where direct conservation action or control of the invader are needed, but can also highlight gaps in models where better information is needed. Furthermore, this method could be used to compare the impacts of different pest species and prioritise control approaches given limited resources.

## ACKNOWLEDGEMENTS

Thanks to those who provided additional distribution data of Asian paper wasps, especially John Early, Phil Sirvid, and Richard Toft. Thanks to Catalina Amaya-Perilla and Tom Saunders for assistance with field surveys.

### Funding

This work was supported by the Ministry for Science and Innovation through funding of the Managing Invasive Weeds, Pests and Diseases Portfolio (project #CF1112-93-02). The funders had no role in study design, data collection and analysis, decision to publish, or preparation of the manuscript.

### Grant Disclosures

The following grant information was disclosed by the authors:
Ministry for Science and Innovation through funding of the Managing Invasive Weeds, Pests and Diseases Portfolio: #CF1112-93-02.

### Competing Interests

Darren Ward and Fraser Morgan are employees of Landcare Research. The authors declare there are no competing interests.

### Author Contributions

- Darren Ward and Fraser Morgan conceived and designed the experiments, performed the experiments, analyzed the data, contributed reagents/materials/analysis tools, wrote the paper, prepared figures and/or tables, reviewed drafts of the paper.

### Field Study Permissions

The following information was supplied relating to field study approvals (i.e., approving body and any reference numbers):

Permission for field sampling was obtained from the Auckland Council (#SS42).

### Supplemental Information

Supplemental information for this article can be found online at http://dx.doi.org/10.7717/peerj.435.

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
