# Peer review of "Modelling the impacts of an invasive species across landscapes: a step-wise approach"

_PeerJ, doi:10.7717/peerj.435_

## Round 0.1 · original submission · Minor Revisions

While I came to your manuscript a little late in the game, I have thoroughly reviewed it and the reviewers comments. with revision I think your paper will be suitable for PeerJ. At PeerJ (like many other journals) the distinction between major and minor revision is rather subjective, although papers with major revision may require a second peer review. While revision is needed, I don't see an additional peer review being necessary.

Please respond to the specific points raised by both reviewers in your revision and/or your letter to me. I think a most important issue is to address your "hierarchical" model. I think reviewer one's difficulty was in distinguishing between the specifics of what you did, and statistical models. To address this issue, I think making your procedures more explicit, and specifically distinguishing what you did (a step-wise selection process if I understood you correctly) from other modeling approaches that might be termed hierarchical. On a related point, would you please include more information, such as sentence or two description, regarding your analysis software MaxEnt, as well as details on publisher, location, etc. Also, although neither reviewer noted this point, I think you need to explain how you selected data for your model versus data for validation. I assume you used some form of stratified random procedure (in that you needed different habitats, but could use random data within habitats) but please be detailed on this process. And, please explain why you used a 10C base temperature for your degree day estimate (I think "degree day" is the preferred form rather than "growing degree day", and how were degree days calculated (simple daily avg minus base?).

On a stylistic point, I am heartily against the use of abbreviations in scientific writing unless the abbreviations are standard and well known. My justification for this opinion, is that when a reader encounters an abbreviation that is not immediately recognized, they stop, recall the meaning, and only then continue with the sentence. Consequently, abbreviations can become a serious barrier to understanding and clarity. While I won't require you to change your abbreviations, I hope you will replace them with text. None of the abbreviations for your habitats represent standards so your readers will either need great memories or an abbreviation key as they read (rather like the list of characters I had to refer to the first time I read War and Peace). Anyway, please give this modification serious thought.

Reviewer 1 ·

Basic reporting

The study seeks to spatially quantify “impact” from the Asian paper wasp in New Zealand. The paper is reasonably well-written, although it could use some editing. The introduction provides reasonable background on the paper wasp, but it is a bit lacking in some necessary details. Authors should provide a definition of MaxEnt and a bit of background on bioclimatic niche envelopes. See specific edits in the text.

Experimental design

The authors refer to the “hierarchical” design of the methods. I find this a misnomer as they are not using any form of nested analysis. Instead, they are using a systematic string of assumptions to move from known occupancy to quantifying potential impact. The logic behind the steps is reasonable, but not hierarchical. They should change the name of the paper and the use of the term “hierarchical” or convince me that this is somehow a nested set of steps. I would call this “additive” or “systematic” or something else besides hierarchical. They may be trying to say hierarchy from the perspective of scaling from known occupancy locally to NZ-wide estimate of impact, but it still doesn’t work for me.

Validity of the findings

The authors can do much more to discuss the caveats of their approach. Further, they have made no effort to assess sensitivity to various assumptions and thresholds used in the report. Without this, this reads more like a master’s level paper than a scientific report. The “impacts” estimated are vague and unitless. I think Figure 5 shows number of prey items. But what does this mean in regard to actual impact? This must be different for urban vs. agricultural areas, etc. What does this actually mean for management of the paper wasp? I am neither convinced that the results are robust nor that they provide sufficient information to guide future science or management. So, what is the point of all this?

Additional comments

I think this is publishable, but it needs to be reframed in two ways: 1) more robust science with sensitivity analyses, and 2) more robust conclusions about what it all means.

Reviewer 2 ·

Basic reporting

The article appears to meet PEER J's standards.

Experimental design

The article meets the journal's standards.

Validity of the findings

The results and conclusions seem robust based on the methods and assumptions of the models. See my comment below about using absence data as well as occurrence data.

Additional comments

In this paper, the authors use modeling approaches to estimate a number of characteristics of an introduced species of Polistes to New Zealand including: its potential range, density, and impact through predation. Overall, I found the paper to be novel and the data worth publishing. I had one main comments and a few minor suggestions.

Regarding the data construction based on occurrence data. The model can be improved by also including absence data from surveys that did not find the wasps. Is such data available (for example from the density estimate data collection part of this paper, other surveys, or the literature)?

Specific comments:

Line 10 “the habitats of” change to “the distribution of”?

Line 13 delete “However”

Line 13 change to “are” analyses.

Line 23 “Roy HE” – author initials should be deleted in this citation.

Line 23. You should add a sentence or two about the biology of Polistes wasps and a little background about their role as invaders here.

Line 39 these final two paragraphs can be combined.

Figure 1 and legend. Each inset box should be clearly labeled and matched with main map.

---

## Round 0.2 · accepted · Accept

Thanks for your timely revision and rebuttal letter. I think you have addressed all the key points raised and the paper is suitable for publication. I think it reads better without the abbreviations so thanks for making that change in particular.